# Incidence of Guillain-Barré Syndrome Is Not Associated with Influenza Vaccination in the Elderly

**DOI:** 10.3390/vaccines8030431

**Published:** 2020-07-31

**Authors:** Hankil Lee, Hye-Young Kang, Sun-Young Jung, Young-Mock Lee

**Affiliations:** 1CONNECT-AI research center, Yonsei University Health System, Seoul 03722, Korea; hankil912@gmail.com; 2College of Pharmacy, Yonsei Institute of Pharmaceutical Sciences, Yonsei University, Incheon 21983, Korea; hykang2@yonsei.ac.kr; 3College of Pharmacy, Chung-Ang University, Seoul 06974, Korea; jsyoung01@gmail.com; 4Department of Pediatrics, Gangnam Severance Hospital, Yonsei University College of Medicine, Seoul 06273, Korea

**Keywords:** Guillain-Barré syndrome, influenza vaccine, incidence, self-controlled case series

## Abstract

We aimed to analyze the incidence of Guillain-Barré syndrome (GBS) and its association with influenza vaccination (IV) in the elderly population. This study included 2470 patients hospitalized with GBS (G61.0) between 2014 and 2016 based on the Korean National Health Insurance Service (NHIS) claims data. We reviewed every medical claim in the 42 days preceding GBS diagnosis looking for precedent causes of GBS. To assess the relationship between IV and the development of GBS, data from the NHIS and the National Vaccination Registry were combined and analyzed. Using a self-controlled case series (SCCS) approach, we calculated the incidence rate ratio by setting the risk period as 42 days following vaccination. The annual background incidence of GBS was estimated at 4.15 per 100,000 persons. More than half of the patients with newly developed GBS had a previous infection or surgery. The incidence of GBS within 42 days of IV was estimated at 0.32 per 100,000 vaccinated persons. SCCS analysis showed that the risk of GBS was not significantly higher. While GBS can potentially develop from various infections, no association was found between GBS and IV. These results will contribute to developing an evidence-based vaccine policy that includes a clear causality assessment of adverse events.

## 1. Introduction

Influenza is a common infectious disease considered to be socially and economically burdensome, especially because of its frequent complications [1,2]. Vaccination is the most effective method for preventing influenza [3,4], and it is especially recommended for high-risk groups [5]. In Korea, influenza vaccination has been included in the National Immunization Program (NIP) since 1997 [6], and its coverage among the elderly population reached levels above 85% in 2018, which is the highest coverage rate in the world [7]. In comparison, Japan reported a coverage rate of 48% in 2017, and China reported a rate of 38.7% in Beijing in 2012 [8].

Importantly, concerns about adverse events following immunization (AEFI) are increasing. One AEFI of particular interest, which may occur after influenza vaccination, is Guillain-Barré syndrome (GBS) [9]. GBS is a rare but severe type of peripheral neuropathy [10]. The incidence of GBS increases with age and is higher in men than in women. For example, one meta-analysis found that, the incidence of GBS among people aged 60–69 and 70–79 years was 1.85 and 2.22 per 100,000 person years, respectively; whereas, the incidence among people aged 20–29 years was 0.90 per 100,000 person years [11]. Although a variety of infections have been reported as causes of GBS [12,13], influenza vaccination stands out as it has been repeatedly mentioned as possibly causative of GBS [14,15].

Despite the efforts of the WHO vaccine safety committee and the Institute of Medicine, the potential mechanism of action by which influenza vaccination could cause GBS is unknown [16,17]. Nevertheless, a recent meta-analysis found a significant association between influenza vaccination and the incidence of GBS [18]. However, the results varied depending on the study year, study design, data source, and region. Furthermore, Kwong et al. (2013) reported that GBS was more associated with influenza itself than with influenza vaccination [19], insisting instead that influenza vaccination could decrease GBS incidence.

Notably, most of the studies to date have been conducted in Western countries, while evidence concerning the association and basic epidemiologic information about GBS in Asian populations is lacking. Moreover, there has been recent advancements as well as diversification in epidemiological methodology. In particular, the self-controlled case series (SCCS) method is now widely recognized as a valid approach to assess the association between vaccination and AEFI, while minimizing confounders associated with vaccinated individuals, through the use of cases themselves as controls [20,21,22].

Therefore, we conducted a nationwide study to examine the epidemiologic characteristics of GBS in the Korean elderly population. Since the development of GBS increases with age and the Korean NIP offers free vaccination against influenza for the elderly, our study focused on the population aged 65 years or older. Specifically, we aimed to: (1) estimate the incidence rate of GBS regardless of vaccination; (2) estimate the incidence rate of GBS among vaccinated people; and (3) examine pre-existing conditions that could lead to GBS. In addition, we aimed to (4) assess the association between influenza vaccination and GBS development following vaccination by merging the National Vaccination Registry data and the National Health Insurance Service (NHIS) claims data. These results will contribute to the development of an evidence-based vaccine policy, and help the National Vaccine Injury Compensation program, which is in urgent need of a clear causality assessment for AEFI.

## 2. Materials and Methods

### 2.1. Data Sources

To examine the epidemiologic characteristics of GBS, we first estimated its incidence rate regardless of vaccination. We used the claims data from the NHIS to examine any pre-existing conditions that may lead to the development of GBS. The NHIS claims data covers the entire population of Korea and provides selected demographic information (i.e., sex, age, residence area, economic status, and disability) of the enrollees, as well as records of medical interventions including diagnoses, procedures, and prescribed drugs [23].

To estimate the incidence rate of GBS among vaccinated individuals and to assess the association between influenza vaccination and GBS, we used two national datasets: the vaccination registry established by the Korea Centers for Diseases Control and Prevention (KCDC) [24] and the NHIS claims data. The KCDC keeps a vaccination registry for all vaccines covered by the NIP, in which physicians have to register the vaccination information when they inoculate patients within the NIP. The vaccination registry data includes demographic characteristics of the vaccinated subjects (i.e., sex, age, and residence area), and information about the vaccine (i.e., date of vaccination, type of vaccine, and dose). We merged the vaccination registry data from 2014 to 2016 with the NHIS claims data from 2013 to 2017 using de-identified individual identification numbers. We covered two seasons of influenza vaccination: the 2014–2015 season, corresponding to the period from July 2014 to June 2015, and the 2015–2016 season, corresponding to the period from July 2015 to June 2016.

This study protocol was approved by the Institutional Review Board of Yonsei University Gangnam Severance Hospital, Seoul, Republic of Korea (IRB No. 3-2018-0116). The requirement to obtain informed consent was waived due to the untraceable and de-identified nature of the secondary claims data. This study was performed in accordance with the ethical guidelines of the Declaration of Helsinki of the World Medical Association. The Strengthening the Reporting of Observational Studies in Epidemiology (STROBE) statement was used for reporting [25].

### 2.2. Guillain-Barré Syndrome Case Ascertainment

A new GBS onset was defined as a case of hospitalization with a primary diagnosis of GBS (international classification of disease (ICD)-10th code G61.0) and identified as follows. For each one-year study period (July 2014–June 2015 or July 2015–June 2016), we identified the patients having hospitalization records with a primary diagnosis of GBS, aged 65 years or older at the time of hospitalization. To define a new occurrence of GBS, we set the year before the earliest hospitalization with GBS in the study period as a washout period. We excluded patients if they had records of outpatient visits or hospitalization with a diagnosis of GBS or polyneuropathy (ICD-10th codes G61.8, G61.9, G62.8, or G62.9) during the washout period [19]. Patients who had outpatient visits due to GBS, or outpatient visits or hospital admission due to polyneuropathy during the month before the hospitalization with GBS, were considered to have received pre-treatments associated with hospitalization due to GBS, and therefore included as incident GBS cases. Figure 1 shows how the incident GBS cases were identified from the NHIS claims data.

### 2.3. Data Analysis

We first estimated the background incidence of GBS, which occurs regardless of influenza vaccination. Next, the incidence of GBS that occurred within 42 and 90 days after influenza vaccination was calculated. Then, we analyzed the frequency of potential causes of GBS in the medical history within 42 days of the occurrence date of GBS. Finally, using the SCCS method, the relative incidence of GBS in the exposed period compared to the unexposed period of the influenza vaccine was calculated for those with GBS who had received influenza vaccination.

#### 2.3.1. Incidence Rate of GBS

The incidence rate of GBS was estimated based on two measurements for each influenza season: one was the background incidence rate of GBS regardless of vaccination and the other one was the incidence rate of GBS following influenza vaccination within the risk period of 42 days (6 weeks) or 90 days (3 months). The background incidence rate of GBS per 100,000 elderly person years was calculated by dividing the number of new patients with GBS per year by the size of the elderly population for the corresponding year, obtained from Statistics of Korea [26]. In order to estimate the incidence rate of GBS following influenza vaccination, we selected patients who had a history of influenza vaccination among the new patients with GBS, from which we extracted those who developed GBS within 42 days (6 weeks) or 90 days (3 months) after the influenza vaccination [19,27]. Thus, the incidence rate of GBS following the influenza vaccination per 100,000 elderly persons was calculated by dividing the number of these patients by the total number of elderly people vaccinated against influenza.

Background incidence rate of GBS among the elderly per year = (Number of elderly patients with new GBS/total size of elderly population) × 100,000.

Incidence rate of GBS within 42 days following influenza vaccination among the elderly = (Number of elderly patients vaccinated against influenza and experiencing GBS within 42 days following vaccination/total number of elderly people vaccinated against influenza) × 100,000.

Incidence rate of GBS within 90 days following influenza vaccination among the elderly = (number of elderly patients vaccinated against influenza and experiencing GBS within 90 days following vaccination/total number of elderly people vaccinated against influenza) × 100,000.

Demographic and health care utilization characteristics of the incident GBS patients were examined, including gender, age, type of national health security program, month of hospital admission, specialty department in which they were treated when hospitalized, type of healthcare institution utilized, and registration in the rare-disease registry.

#### 2.3.2. Precedent Medical Events

Up to 70% of GBS cases are known to be caused by precedent infections including influenza virus infection [28,29]. Additionally, studies have shown that surgery and influenza vaccination can cause GBS [30]. The leading causes of GBS are believed to be cytomegalovirus and upper respiratory tract or gastrointestinal tract infections [12,13]. We analyzed the frequency distribution of precedent infections, surgeries, or influenza vaccination, observed within 42 days of the date of admission with new-onset GBS. The precedent infections examined were selected based on previous literature as well as consultation with a panel of ten clinical experts in Korea [12,13,30,31,32]. Appendix A shows the ICD-10th diagnostic codes for the infections included in our analysis.

#### 2.3.3. Association between Influenza Vaccination and GBS

To assess the association between the influenza vaccination and the development of GBS, we employed the SCCS study design. The SCCS approach sets the risk period and compares the number of events of interest between the risk period and the control (non-risk or baseline) period within the same study subject [20]. The risk period is defined as the time during which immunization has a high probability of causing AEFI. It is generally set based on biological plausibility and half-life exposure, but also by empirical or customary consensus. It is also possible to define one or more risk periods [20,21,22].

Based on previous literature and consultation with clinical experts, we set the risk period for our analysis at 42 days after influenza immunization, meaning that the influenza vaccination could cause GBS within the 42 days following vaccination [33,34]. The risk period was further divided into the sub-periods ≤3, 4–7, 8–14, and 15–42 days to analyze the causal relationship between vaccination and GBS according to different risk periods (Figure 2). Additionally, the period of 43–90 days was set as an additional risk period to examine the possibility of GBS development even after a considerable time gap following influenza vaccination. The control period was defined as having no relationship with influenza vaccination, namely the period before or more than 90 days after vaccination.

Using conditional Poisson regression analysis, we estimated the incidence rate ratio (IRR) comparing the incidence of GBS in the risk period with that in the control period. All data analyses were performed using the SAS program (9.4 version, Cary, NC, USA). The SAS code used to perform SCCS was adapted from the instructions and macrocode provided by The Open University (http://statistics.open.ac.uk/sccs).

### 2.4. Data Availability Statement

Data are available upon request due to data protection guidelines laid down by the Korean data privacy regulation. Data access requests may be made to the Institutional Review Board of Yonsei University Gangnam Severance Hospital at gsirb@yuhs.ac.

## 3. Results

### 3.1. The Background Incidence Rate and Characteristics of GBS Patients

Out of 1191 patients hospitalized with GBS from July 2014 to June 2015, we identified 262 patients aged 65 years or older with new-onset GBS (Figure 1). In addition, out of 1279 patients hospitalized with GBS from July 2015 to June 2016, 271 patients aged 65 years or older with new-onset GBS were identified. The annual incidence rate of GBS among the population aged 65 years or older was calculated to be 4.16 per 100,000 persons in the 2014–2015 season and 4.14 per 100,000 persons in the 2015–2016 season. The incidence rate was higher among men than women (Table 1).

More than half of the elderly patients with new-onset GBS were aged 65–74 years (59.1%) and no seasonality was observed in the development of GBS (Table 2). More than three quarters of the patients with new-onset GBS were diagnosed in the department of neurology, and the majority of patients were hospitalized in general or tertiary-care hospitals.

### 3.2. Prevalence of Precedent Medical Events before the Development of Guillain-Barré Syndrome

As shown in Table 3, more than half of the patients with new-onset GBS had precedent medical events such as infections, surgery, or influenza vaccination within 6 weeks prior to onset, which may be potential causes for the development of GBS. Upper respiratory infections showed the highest prevalence in GBS patients (more than 40%), followed by gastrointestinal (GI) infection (more than 10%). Among pathogens, influenza virus was the most frequently reported (9.57%), followed by herpes zoster virus (2.25%), and herpes simplex virus (0.95%). However, only one case of Campylobacter jejuni, which is known to cause GBS [12,13], was reported, and no patient had been infected with the cytomegalovirus or the Epstein–Barr virus. Approximately 5–7% of the GBS patients had received the influenza vaccine within 6 weeks of hospitalization with GBS.

### 3.3. Incidence Rate of GBS among the Elderly Vaccinated against Influenza

Among the incident GBS patients, 320 had a record of influenza vaccination (145 in the 2014–2015 season and 175 in the 2015–2016 season). Among them, 32 experienced the onset of GBS within 42 days following the influenza vaccination, and 74 within 90 days. During the two influenza seasons, the GBS incidence rate within 42 days and 90 days of influenza vaccination among the population aged 65 years or older was 0.32 and 0.73 per 100,000 persons vaccinated against influenza, respectively (Table 4).

### 3.4. Association between Influenza Vaccination and GBS Development

Using the SCCS approach, we estimated the IRR of GBS for each risk period to assess the association between influenza vaccination and the development of GBS. The IRR for the risk periods of 4–7 days, 8–14 days, 15–42 days, and 43–90 days were 0.83 (95% confidence interval [CI]: 0.26–2.58), 0.94 (95% CI: 0.42–2.12), 0.86 (95% CI: 0.56–1.34), and 0.89 (95% CI: 0.64–1.25), respectively (Table 5). None of the IRRs were statistically significant, suggesting the lack of association between influenza vaccination and the development of GBS. Subgroup analysis by sex and age consistently showed non-significant IRRs (Appendix A).

## 4. Discussion

This epidemiological study examined the incidence rate of GBS and assessed the association between influenza vaccination and the development of GBS in the elderly population. We estimated the background annual incidence rate of GBS at 4.14–4.16 per 100,000 elderly persons. The incidence was higher in men than in women, with an incidence ratio of 1.5 without seasonality. In addition, the incidence rates of GBS within 42 and 90 days following influenza vaccination were estimated as 0.32 and 0.72 per 100,000 elderly persons vaccinated, respectively. In this study, precedent infection or a history of surgery within 42 days before the incidence of GBS were found in more than half of the patients. The most common infections were upper respiratory infection and GI infection, which are very common in the general population. Using the SCCS approach, we found that the IRRs of GBS for each risk period were not statistically significant.

The incidence rate of GBS (4.14–4.16 per 100,000 elderly persons) in this study is higher than that previously reported in a meta-analysis of an elderly population in their 70s (2.22 per 100,000 elderly persons) [11]. Importantly, in Korea, to input the diagnostic code of GBS for a reimbursement, invasive tests such as lumbar puncture and electrodiagnostic tests are required. Because of this, it is likely that GBS is more easily detected in Korea, which has relatively good health accessibility given the universal health insurance system operated by the government [35]. The incidence ratio of males to females in this study (1.5) is similar to the ratio of 1.8 reported in previous studies [36]. Male predominance in the incidence of GBS is a well-known phenomenon, but the cause is still unclear [11]. In this study, the incidence rates of GBS after influenza vaccination within 42 days were higher than that reported by the WHO GACVS in all age groups (0.1–0.2 per 100,000 persons who received influenza vaccination) [16], but lower than the GBS incidence in the elderly aged 65 years and older regardless of influenza vaccination in the present study (4.1–4.2 per 100,000 person years).

In this study, more than half of the patients with a new GBS had precedent infection or a history of surgery within 42 days prior to the incidence of GBS. This suggests that a very cautious approach is required to determine the causes of GBS. It is noteworthy that approximately 5–7% of the patients with GBS were vaccinated against influenza, while approximately 10% were infected with the influenza virus itself within 42 days before hospitalization with GBS. Although this study does not yield an incidence rate for each precedent cause of GBS, it suggests that additional consideration is needed given the magnitude of the risk that the influenza vaccine and influenza virus itself confer to the incidence of GBS. For example, Kwong et al. (2013) reported that influenza infection (relative risk 15.81; 95% CIs 10.28–24.32) increased the risk of GBS more than the influenza vaccine (relative risk 1.52; 95% CIs 1.17–1.99) [19]. Therefore, in order to balance the benefits and risks of influenza vaccination, it is important to establish policies for influenza vaccination while considering the management of GBS incidence.

To assess the casual association between the incidence of GBS and influenza vaccination, we integrated the national vaccination registry and NHIS claims data for the entire elderly population. Using the SCCS method, we found no significant causal association. However, this association remains controversial. As a result of combining 22 research findings from the latest studies by Arias et al. (2015), the relative risk of GBS occurrence after seasonal influenza vaccination was 1.22 (95% CI: 1.01–1.48) [18]. However, among the 22 studies, the relative risk of the four that used the SCCS method like the present study was 1.14 (95% CI, 0.78–1.66), which was not statistically significant and thereby suggests a lack of a significant association of GBS with influenza vaccination, in agreement with the results of the present study [18]. The SCCS method has an advantage in controlling confounding variables that are not measured or not time-dependent in cases where it is difficult to define a non-exposed group, such as people not vaccinated with influenza, due to its high coverage rate. The cases contain their own controls, allowing us to compare the number of events that occurred in the risk-period with those occurring in the control period. The IRR calculated from the SCCS analysis represents the relative incidence of the risk periods with respect to the control period. Therefore, this value does not rely on the comparison of incidence in people who have and have not been vaccinated.

The following points must be noted in interpreting the results of this study. First, we used hospitalization status and disease codes/order in the health insurance claims data instead of checking the medical charts of treatment in defining GBS patients. Therefore, the number of patients may have been underestimated or overestimated, depending on the accuracy of diagnosis. However, the code diagnosing of GBS is G61.0, which is a very specific code, and according to the experts’ panel the approach of selecting inpatients diagnosed with GBS as the primary diagnosis of the study subjects was deemed adequate. Moreover, previous studies conducted in other countries also selected inpatients with G61.0 as the definition of GBS cases [19]. For example, a study in Korea that defined GBS by reviewing medical records also reported that the G61.0 code was recorded in the primary diagnosis or secondary diagnosis [37]. This implies that defining the inpatients with G61.0 as the primary diagnosis of GBS cases is a valid approach, since the G61.0 code is expected to have a high positive predictive value (PPV). In an algorithmic approach to the diagnosis of GBS based on claims data, an algorithm including a primary inpatient GBS code and a neurologist visit associated with the GBS code gave the highest PPV of 0.70 [38]. Second, it is difficult to specify the GBS onset date, because of the nature of the disease. Thus, the onset date is considered to coincide with the date of admission when interpreting the IRR. To overcome this limitation, we defined an extra risk period (43–90 days) and sub-divided the risk period into the sub-periods ≤3, 4–7, 8–14, and 15–42 days. Third, the absolute incidence of GBS following influenza vaccination cannot be calculated in the SCCS method. This research design selects patients with GBS and patients with records of influenza vaccination as the study subjects, and thus can efficiently estimate the IRR, but does not estimate the incidence of GBS among all influenza-vaccinated patients. However, efforts were made to overcome the limitations of the SCCS method by also estimating the GBS incidence rate within the risk period following influenza vaccination.

The analysis of the total number of GBS cases diagnosed in the Korean elderly population showed that the background incidence of GBS is slightly higher than in other countries. However, as shown in our results, more than half of the GBS patients had a history of common infections or surgery and influenza vaccination was considered one of such possible causes. Thus, to make the evidence-based vaccine policy that minimizes GBS as AEFI, the correct diagnosing of GBS as well as a thorough review of a patient’s medical history are critical. Additionally, the statistically insignificant IRR results of the GBS incidence after influenza vaccination compared to unexposed risk period are meaningful for the elderly population. This result can be specifically applied to the process of decision making in the Korean National Vaccine Injury Compensation Program.

In order to accurately diagnose GBS and implement a comprehensive vaccine policy, the government should sustainably perform a long-term pharmacoepidemiologic study and establish an active monitoring system that enables the tracking of vaccination history and AEFI incidence in a unified database. In addition, clinicians should be knowledgeable about patients’ health conditions such as a history of infections or surgery as well as vaccinating when diagnosing GBS.

## Figures and Tables

**Figure 1 vaccines-08-00431-f001:**
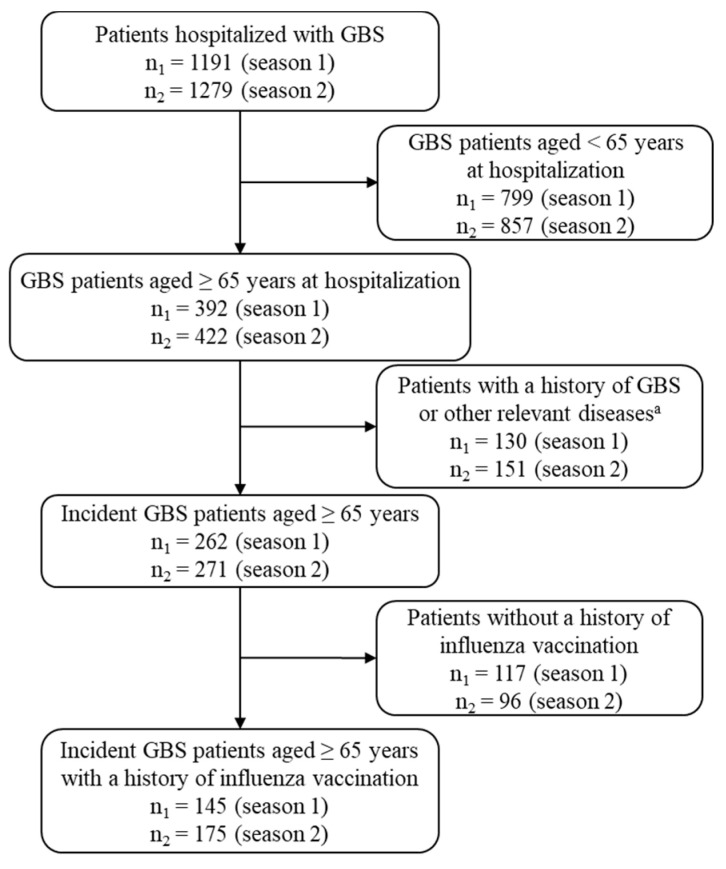
Flow chart of the study population selection. (^a^ Excluding patients with a history of GBS or inflammatory demyelinating polyneuropathy during the past 1 year from the time of the first GBS admission in the study period; Season 1: July 2014–June 2015; Season 2: July 2015–June 2016).

**Figure 2 vaccines-08-00431-f002:**
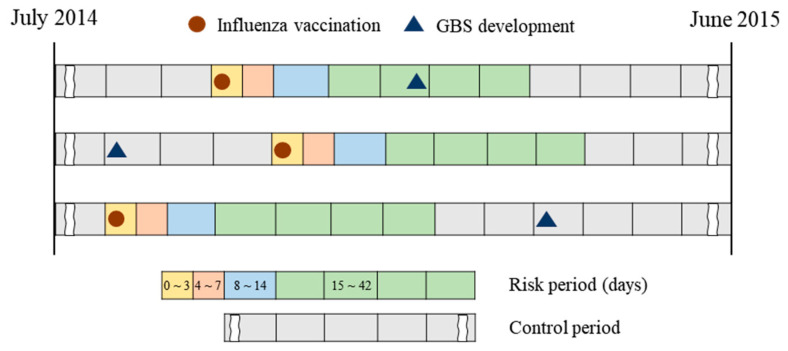
The concept of the self-controlled case series approach in the 2014–2015 season.

**Table 1 vaccines-08-00431-t001:** Background incidence of Guillain-Barré syndrome in the Korean elderly population.

Incidence	Overall (July 2014–June 2016)		2014–2015 Season (July 2014–June 2015)		2015–2016 Season (July 2015–June 2016)	
	No. of Patients	Incidence per 100,000	No. of Patients	Incidence per 100,000	No. of Patients	Incidence per 100,000
Total	533	4.15	262	4.16	271	4.14
Sex	-	-	-	-	-	-
Male	278	5.18	140	5.34	138	5.02
Female	255	3.41	122	3.32	133	3.50
Age (years)	-	-	-	-	-	-
65–74	315	4.01	158	4.07	157	3.94
75–84	192	4.43	95	4.50	97	4.36
≥85	26	2.37	9	1.70	17	2.99

**Table 2 vaccines-08-00431-t002:** Demographic and healthcare utilization characteristics of the elderly patients with incident Guillain-Barré syndrome.

Characteristics	Overall (July 2014–June 2016) *n* = 533		2014–2015 Season (July 2014–June 2015) *n* = 262		2015–2016 Season (July 2015–June 2016) *n* = 271	
	n	%	n	%	n	%
Sex	-	-	-	-	-	-
Male	278	52.16	140	53.44	138	50.92
Female	255	47.84	122	46.56	133	49.08
Age (years)	-	-	-	-	-	-
65–74	315	59.10	158	60.31	157	57.93
75–84	192	36.02	95	36.26	97	35.79
≥85	26	4.88	9	3.44	17	6.27
Type of national health security program	-	-	-	-	-	-
National Health Insurance	497	93.25	248	94.66	249	91.88
Medical Aid	36	6.75	14	5.34	22	8.12
Month of hospital admission		-	-	-	-	-
July	56	10.51	25	9.54	31	11.44
August	49	9.19	27	10.31	22	8.12
September	40	7.50	18	6.87	22	8.12
October	28	5.25	12	4.58	16	5.90
November	30	5.63	10	3.82	20	7.38
December	35	6.57	22	8.40	13	4.80
January	43	8.07	25	9.54	18	6.64
February	37	6.94	16	6.11	21	7.75
March	58	10.88	29	11.07	29	10.70
April	50	9.38	21	8.02	29	10.70
May	62	11.63	35	13.36	27	9.96
June	45	8.44	22	8.40	23	8.49
Physician specialty treating GBS	-	-	-	-	-	-
Neurology	413	77.49	202	77.10	211	77.86
Rehabilitation	35	6.57	20	7.63	15	5.54
Internal medicine	36	6.75	16	6.11	20	7.38
Neurosurgery	25	4.69	10	3.82	15	5.54
Others	24	4.50	14	5.34	10	3.69
Type of healthcare institutions treating GBS	-	-	-	-	-	-
Tertiary-care hospital	302	56.66	151	57.63	151	55.72
General hospital	111	20.83	100	38.17	11	4.06
Hospital	13	2.44	7	2.67	6	2.21
Long-term care hospital	7	1.31	4	1.53	3	1.11
Rare disease registry	-	-	-	-	-	-
Yes	435	81.61	208	79.39	227	83.76
No	98	18.39	54	20.61	44	16.24

(GBS: Guillain-Barré syndrome).

**Table 3 vaccines-08-00431-t003:** Precedent medical events during the 6 weeks before the onset of Guillain-Barré syndrome.

Category		Overall (July 2014–June 2016) *n* = 533		2014–2015 Season (July 2014–June 2015) *n* = 262		2015–2016 Season (July 2015–June 2016) *n* = 271	
		*n*	%	*n*	%	*n*	%
**Any infection/surgery/influenza vaccine**		312	58.54	154	58.78	158	58.30
Anatomical infection site	Upper respiratory	225	42.21	110	41.98	115	42.44
	Lower respiratory	1	0.19	0	0.00	1	0.37
	Gastro-intestinal	65	12.20	31	11.83	34	12.55
Pathogen	Malaria	0	0.00	0	0.00	0	0.00
	Tsutsugamushi	3	0.56	0	0.00	3	1.11
	Campylobacter	1	0.19	1	0.38	0	0.00
	Cytomegalovirus	0	0.00	0	0.00	0	0.00
	Epstein-Barr virus	0	0.00	0	0.00	0	0.00
	Herpes simplex virus	5	0.94	3	1.15	2	0.74
	Varicella (Chickenpox)	0	0.00	0	0.00	0	0.00
	Herpes zoster virus	12	2.25	4	1.53	8	2.95
	Mycoplasma	0	0.00	0	0.00	0	0.00
	Hemophilus influenzae	0	0.00	0	0.00	0	0.00
	Influenza virus	51	9.57	26	9.92	25	9.23
	Parainfluenza virus	0	0.00	0	0.00	0	0.00
	Other viruses	0	0.00	0	0.00	0	0.00
	Other infections	0	0.00	0	0.00	0	0.00
Surgery		44	8.26	23	8.78	21	7.72
Influenza vaccine (trivalent) *		32	5.82	13	4.96	19	7.01

* All influenza vaccines to be studied are trivalent vaccines, and the compositions of the vaccines are as follows: In the 2014–2015 season: An A/California/7/2009 (H1N1)pdm09-like virus; an A/Texas/50/2012 (H3N2)-like virus; and a B/Massachusetts/2/2012-like virus. In the 2015–2016 season: An A/California/7/2009 (H1N1)pdm09-like virus; an A/Switzerland/9715293/2013 (H3N2)-like virus; and a B/Phuket/3073/2013-like virus [5].

**Table 4 vaccines-08-00431-t004:** Incidence rate of Guillain-Barré syndrome among those vaccinated against influenza.

Incidence	Overall (July 2014–June 2016)		2014–2015 Season (July 2014–June 2015)		2015–2016 Season (July 2015–June 2016)	
	No. of Patients	Incidence per 100,000 Vaccinated	No. of Patients	Incidence per 100,000 Vaccinated	No. of Patients	Incidence per 100,000 Vaccinated
Within 42 days post vaccination						
Total	32	0.32	13	0.28	19	0.35
Male	16	0.38	6	0.31	10	0.45
Female	16	0.27	7	0.25	9	0.28
Within 90 days post vaccination						
Total	74	0.73	39	0.83	35	0.65
Male	33	0.79	20	1.03	13	0.58
Female	41	0.69	19	0.69	22	0.69

**Table 5 vaccines-08-00431-t005:** Results of conditional Poisson regression analysis to assess the association between influenza vaccination and Guillain-Barré syndrome following vaccination.

Risk Period (Days Following Vaccination)	Total (*n* = 320)				2014–2015 Season (*n* = 145)				2015–2016 Season (*n* = 175)			
	*n* (%)	IRR	95% CI		*n* (%)	IRR	95% CI		*n* (%)	IRR	95% CI	
			Lower Limit	Upper Limit			Lower Limit	Upper Limit			Lower Limit	Upper Limit
**Baseline (Before influenza vaccination or after vaccination of 90 days)**	251 (78.44)	1.00	-	-	110 (75.86)	1.00	-	-	141 (80.57)	1.00	-	-
0–3	0 (0)	-	-	-	0 (0)	-	-	-	0 (0)	-	-	-
4–7	3 (0.94)	0.83	0.26	2.58	1 (0.69)	0.63	0.09	4.48	2 (1.14)	0.98	0.24	3.97
8–14	6 (1.88)	0.94	0.42	2.12	1 (0.69)	0.36	0.05	2.56	5 (2.86)	1.40	0.58	3.42
15–42	21 (6.56)	0.86	0.56	1.34	9 (6.21)	0.80	0.41	1.59	12 (6.86)	0.91	0.52	1.61
43–90	39 (12.19)	0.89	0.64	1.25	24 (16.55)	1.25	0.80	1.94	15 (8.57)	0.61	0.36	1.05

(CI, confidence interval; IRR, incidence rate ratio).

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
