# Peer review of "Incidence of Guillain-Barré Syndrome Is Not Associated with Influenza Vaccination in the Elderly"

_vaccines, 2020, doi:10.3390/vaccines8030431_

Round 1

Reviewer 1 Report

General

Authors explore in their manuscript ‘Incidence of Guillain-Barré Syndrome and the Impact of Influenza Vaccination in South Korea’ the incidence of GBS with the association with influenza vaccination. They used data already collected in South Korea. Side effects of vaccinations are understudied, however some minor work should be done before this manuscript can be published.

Title

The title looks too long, too causal and furthermore too local. ‘Incidence of Guillain-Barré Syndrome is not associated with Influenza Vaccination’; leave out this <in South Korea>, for I think that your finding has a wider meaning than just restricted to S Korea; you can think it over whether you want to extend the title with <in elderly>.

Abstract

Background

Remove <Korean> from the aim (or make clear to me why you want to publish internationally such a local issue).

Methods

Please replace <2016 from> by <2016 based on>

Results

Please change <half the patients> into <half of the patients>

Conclusion

-

Key words

-

Introduction

<In Korea> please go into the regional situation, as readers might know how Korea is doing, compared to e.g. Japan or China.

Please change <GBS is, a rare> into < GBS is, a rare>

Please change <Kwong et al reported> into <Kwong et al. reported>

Methods

Sample

<from 2013 to 2017>; should 2013, as claims will always be later than the vaccination, not be 2015? Please clarify.

Measures

-

Statistical analyses

Please rewrite this section: First, we … . Then, we … . Next, we … . Finally, we … . The readership easier grabs what you did and in which order the Results will be shown.

Was it necessary to seek an approval of an IRB with this study, in which you use data, which are already present? In fact you use two databases.

Results

183-186 This part of the text comes below Table 1 and before Table 2.

Discussion

Please keep in mind the following structure for writing a Discussion:

para1                    start with repeating the research question or paraphrasing it (you did so) + answer this without any comments or interpretation.

para2,3,#             start a new para, 1 topic per para, and start this para with one of your findings (We found; In this study) – which then defines the content of the para (you did so). Relate your finding to earlier published references (you did so). End such a para by saying how your finding changed existing theory (please add).

230        change <a previous meta-analysis9> into <a previous meta-analysis [9]>

236        <On the other hand> you don't show us the one hand.

Strengths and limitations

I would advise you to shorten this part.

(don't start a new discussion on the findings)

Implications

(what do your findings mean for practice/for (health) policy,

what do your findings mean for future research)

Conclusion

Condense; give an answer to the aim of the study and mention your most important implication

Tables, Figures

See Results

References

Be consistent in names, and names of journals

Reviewer 2 Report

Dear Authors, 

the manuscript titled "Incidence of Guillain-Barré syndrome and the impact of influenza vaccination in South Korea" sounds good and fits the journal aim.

However, some revisions are needed.

First of all, please carefully revised the English style and language.

Moreover, it should be good if the Authors might mention the rehabilitation time and if patients were infected during or after the surgical intervention.

It should be great if the Authors could add a table mentioning the kind of vaccinations. Which molecules and excipients are presents for example, in order to have a clear situation. Moreover, I suggest adding information about patients' allergy, if present.

Avoid typos and fix references (page 11, line 230).

Figure 2: should be great if the Authors could add some lines indicating weeks or days in the colored lines.
